# FreqAlign: Frequency-Based Calibration for Mitigating Contextual Hallucinations in Large Language Models

## Abstract

Despite significant progress, large language models (LLMs) continue to exhibit context hallucinations, generating content that either contradicts or fabricates information relative to the retrieved context. To address this issue, we introduce FreqAlign, a frequency alignment-based method designed to mitigate such hallucinations. Our approach consists of two main stages: first, we construct positive and negative sample pairs according to the actual contextual influence on tokens and train a context-aware classifier that evaluates contextual relevance using token frequency information; second, this classifier is employed to recalibrate the importance distribution of original tokens via frequency alignment. A core innovation of FreqAlign is its dual strategy: it suppresses globally frequent tokens that are prone to induce hallucinations, while enhancing semantically salient yet contextually infrequent tokens. We evaluate our method across six widely-used question-answering benchmarks, where it consistently and substantially outperforms strong state-of-the-art baselines. The implementation of FreqAlign, including training scripts, evaluation protocols, and hyperparameter configurations, is publicly available at `https://anonymous.4open.science/r/FreqAlign-BC8E` . This open release enables reproducibility and facilitates future research into frequency-aware prompting and decoding strategies for reducing hallucinations in LLMs. These findings underscore the potential of frequency-aware modeling as a general and effective strategy for reducing hallucinations in LLMs.

## 1 Introduction

Large language models (LLMs) have demonstrated exceptional capabilities in text generation (Naveed et al., 2023), understanding (Huang et al., 2025a), and reasoning (Becker et al., 2024), fueling advances in domains like conversational AI, knowledge-intensive QA, and content creation (Zhang et al., 2025b). However, despite their fluency, LLMs are prone to hallucinations, generating factually incorrect or unsupported outputs that undermine their reliability (Huang et al., 2025a). A particularly harmful form is **context hallucination**, where a model generates content that contradicts the explicit input context it is provided with (Xu et al., 2024). For instance, given a context "Drug A efficacy: 78%, Drug B efficacy: 82%; side effects not significantly different," a model might conclude, "Drug B is more effective with fewer side effects," fabricating an unsupported claim. Such errors erode trust and pose significant risks, especially in high-stakes fields like healthcare where fidelity to evidence is critical (Roustan et al., 2025).

This issue often arises from a **probability misalignment during autoregressive decoding**. Pretraining objectives that favor statistical co-occurrence over factual fidelity can lead models to overemphasize syntactically plausible tokens while underestimating semantically accurate ones (Liang et al., 2025; Zhang et al., 2024). Indeed, analyses show that over 38% of factual errors occur when the correct tokens are ranked beyond the top 50 candidates in the output distribution (Zhang et al., 2025a). To address this, various mitigation methods have been proposed, largely focusing on decoding-time adjustments (Tao et al., 2024; Huang et al., 2024). For example, techniques like contrastive decoding dynamically reshape output probabilities to improve faithfulness.

However, existing decoding strategies face a common limitation: they lack fine-grained lexical control and often fail to distinguish between high-frequency but low-relevance tokens and low-frequency but high-relevance tokens. Recent studies (Holtzman et al., 2019; Shen et al., 2024) have observed that LLMs disproportionately favor globally frequent tokens (e.g., common verbs like "is" or "has"), which can obscure contextually important but rarer terms. While some works have noted the influence of token frequency (Schmied et al., 2025), no prior work has systematically integrated frequency signals into the decoding process to correct the token rank distortion at the core of the problem. This leaves a critical gap in addressing the conflict between a model's internal statistical priors and the provided contextual evidence.

To bridge this gap, we propose FreqAlign, a novel decoding-time paradigm that corrects distortions in the probability distribution through **bidirectional frequency alignment**. Our approach dynamically calibrates token probabilities by simultaneously pursuing two objectives: (i) **Global frequency de-biasing**, which suppresses high-frequency generic tokens using inverse corpus-frequency weighting. (ii) **Contextual frequency enhancement**, which amplifies tokens that are relevant and frequent within the given context (e.g., boosting the rank of "1973" in a time-based QA task). This method offers a distinct advantage over previous techniques by operating directly at the token level to proactively adjust distributions using intrinsic statistical properties of the text.

Our contributions are:

- A framework that corrects token probability distortions by leveraging statistical frequency alignment to enhance context-relevant predictions.

- A two-stage architecture that combines a context-aware classifier with a bidirectional frequency optimization mechanism for decoding.

- Experimental validation showing that FreqAlign outperforms strong decoding baselines (e.g., +3.02% F1 over DAGCD on NaturalQuestionsShort) by directly mitigating probability errors induced by model parameters.

## 2 METHOD

### 2.1 OVERVIEW

FreqAlign mitigates contextual hallucination by using a bidirectional frequency optimization mechanism during generation. This dynamic calibration process rectifies token probability distributions by simultaneously amplifying contextually salient tokens and suppressing generic, high-frequency words. The framework operates through two synergistic processes: (1) contextual frequency reinforcement, which elevates the probability of tokens that appear frequently in the provided evidence (e.g., "$95" in a financial context), and (2) global frequency debiasing, which demotes common tokens (e.g., "important" or "significant") based on their overall corpus prevalence. This approach reshapes the probability landscape to anchor the generated output in the given context, without requiring architectural modifications or task-specific fine-tuning.

The overall framework of FreqAlign is illustrated in Figure 1, which depicts the integration of attention signals, frequency features, and decoding-time adjustments. The system first trains a context-aware classifier using attention ratios and normalized frequency signals; this classifier then guides the bidirectional frequency alignment during inference.

### 2.2 CONTEXT GENERATION TRAINING

#### 2.2.1 CLASSIFIER ARCHITECTURE AND OBJECTIVE.

In the training phase of FreqAlign, we design a context-aware classifier to predict the relevance of tokens within a given context to the correct answer. This classifier uses a logistic regression architecture with L2 regularization. The feature vector $\mathbf{v} \in \mathbb{R}^{d+1}$ for a given token integrates two sources of information:

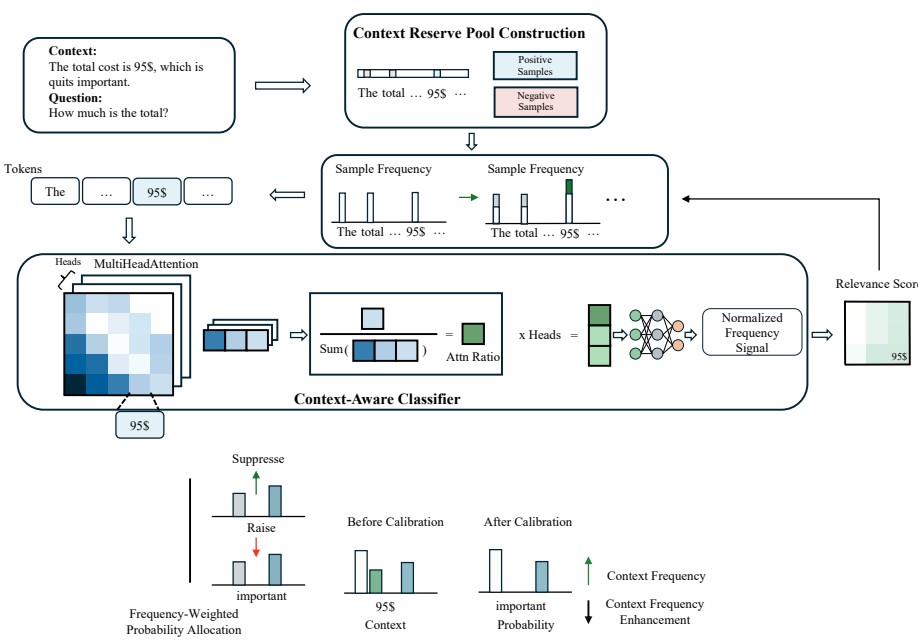

Figure 1: Framework of FreqAlign.

**Multi-head Attention Ratios** ($v_1$ **to** $v_d$)**:** These features are derived from normalized attention weights across the transformer's layers and heads. For each token $w$, we compute:

$$\alpha_{\text{ratio}}^{(i)} = \frac{\alpha^{(i)}}{\sum_{j=1}^{k} \alpha^{(j)}}, \tag{1}$$

where $\alpha^{(i)}$ is the average attention weight of the $i$-th attention head, and $k$ is the total number of heads. This captures the token's relative salience in the model's attention mechanism.

**Normalized Frequency Signal** ($v_{d+1}$)**:** This feature encodes the token's prevalence within the document:

$$f_{\text{norm}}(w) = \frac{\text{count}(w)}{\max_{w' \in \mathcal{V}_{\text{ctx}}} \text{count}(w')}, \tag{2}$$

where $\text{count}(w)$ is the number of times token $w$ appears in the context, and $\mathcal{V}_{\text{ctx}}$ is the set of all words in that context. This feature normalizes the token's frequency against the most frequent token in the context.

The classification objective uses a regularized maximum likelihood estimation framework. The model parameters $\mathbf{w}$ are optimized to minimize the following loss function:

$$\mathcal{L}(\mathbf{w}) = \sum_{i=1}^{N} \ell(y_i, \sigma(\mathbf{w}^{\text{T}} \mathbf{v}_i)) + \lambda \|\mathbf{w}\|_2^2, \tag{3}$$

where $\ell(y, \hat{y})$ is the binary cross-entropy loss, $\sigma$ is the sigmoid activation function, and $\lambda$ is the regularization strength.

The trained classifier produces a relevance score $s(w) = \mathbf{w}^{*\text{T}} \mathbf{v}$, which quantifies the contextual evidence supporting $w$ as a valid component of the answer. This score is central to mitigating hallucination by promoting evidence-based token selection.

### 2.2.2 FREQUENCY-WEIGHTED SAMPLE CONSTRUCTION.

The FreqAlign method uses a frequency-based sampling algorithm to construct its training data. This process is designed to create a balanced and informative dataset by emphasizing high-frequency

answer tokens while down-weighting frequently occurring but irrelevant tokens. The algorithm follows a three-stage process:

**Context Reserve Pool Construction (Stage 1) :** The algorithm first creates separate token pools for positive and negative samples. The positive pool contains "golden" answer tokens drawn from contextually accurate generated responses. The negative pool is populated by filtering context tokens using a dual criterion: first, identifying entities via spaCy's NER pipeline, and second, excluding tokens that already have a high generation probability (top 10). This filtering isolates tokens that are contextually present but have high potential to be part of a hallucinated response.

**Frequency-Weighted Probability Allocation (Stage 2) :** Each token in the pools is assigned a sampling probability proportional to its normalized frequency within the source document. This ensures that the sampling probability is directly related to how common a token is in its original context.

**Hierarchical Resampling (Stage 3):** Finally, hierarchical resampling is performed to construct the final positive and negative training sets. Tokens are selected probabilistically using the frequency-normalized weights calculated in the previous stage. This ensures that high-frequency tokens are well-represented, creating a balanced dataset that reflects the natural distribution of the source data.

This sampling strategy aligns the classifier's decision boundaries with token frequency patterns and helps ensure that the training distribution remains close to the test distribution, enhancing robustness.

## 2.3 FREQUENCY-ALIGNED PROBABILITY ADJUSTMENT

During the inference stage, FreqAlign adjusts the token probability distribution using a bidirectional alignment mechanism to improve semantic consistency and factual accuracy. LLMs often face a challenge where the most probable tokens according to the base model are not factually correct, especially when the right answer consists of lower-ranked tokens. FreqAlign addresses this by using two synergistic adjustments: global frequency de-biasing and context-based frequency enhancement. The final adjustment is a weighted combination of these two signals:

$$\text{combined}_{\text{freq}} = \gamma \cdot f_{\text{ctx}}(w) + (1 - \gamma) \cdot f_{\text{global}}(w), \tag{4}$$

where $\gamma$ is a hyperparameter that balances the influence of the contextual frequency $f_{\text{ctx}}(w)$ and the global frequency $f_{\text{global}}(w)$.

**Global Frequency Debiasing:** This step addresses the tendency of LLMs to favor globally high-frequency words (e.g., "the", "is") that are often irrelevant to the specific task. We apply a penalty to these common words that is inversely proportional to their frequency in a large corpus (calculated using the 'wordfreq' library). This penalty, applied with logarithmic scaling to avoid over-penalizing, effectively reduces the probability of generic tokens, creating space in the top ranks for more context-specific and factually correct tokens. The adjustment is calculated as:

$$P_{\text{debias}}(t) = \frac{1}{\log(\text{frequency}(t) + \epsilon)}, \tag{5}$$

where $P_{\text{debias}}(t)$ is the debiasing factor for token $t$, $\text{frequency}(t)$ is its global corpus frequency, and $\epsilon$ is a small smoothing term.

**Context Frequency Enhancement:** This mechanism increases the probability of tokens that are frequent or significant within the current input context. This helps in two ways: it boosts the chances of selecting rare but important technical terms if they are prominent in the context, and it ensures the generated output reflects the specific vocabulary of the provided evidence. To achieve this, we first calculate a context-specific frequency distribution by counting token occurrences in the input. These counts are then normalized. The final probability is boosted as follows:

$$P_{\text{boosted}}(w) = P_{\text{original}}(w) + \beta \cdot f_{\text{ctx}}(w), \tag{6}$$

where $P_{\text{boosted}}(w)$ is the new probability for token $w$, $\beta$ is a weighting factor, and $f_{\text{ctx}}(w)$ is the normalized contextual frequency. The contextual frequency $f_{\text{ctx}}$ is calculated in real-time by first counting occurrences:

$$\text{count}(w) = \sum_{t \in \mathcal{T}_{\text{ctx}}} \mathbb{I}[t = w], \tag{7}$$

and then normalizing against the most frequent token in the context:

$$f_{\text{ctx}}(w) = \frac{\text{count}(w)}{\max_{w' \in \mathcal{V}_{\text{ctx}}} \text{count}(w')}, \tag{8}$$

This mechanism helps restore correct tokens that might have been suppressed in the original probability distribution by amplifying their importance based on direct contextual evidence.

By combining global debiasing and contextual enhancement, FreqAlign directly adjusts the model's output distribution at each step of the generation process. This proactive approach corrects the probability distribution before a token is selected, reducing the risk of hallucination and ensuring the generated text is both contextually relevant and factually consistent, unlike post-processing methods which can suffer from error accumulation.

## 3 Experiments

We conduct a comprehensive evaluation of FreqAlign on six challenging question-answering datasets to assess its effectiveness in mitigating context hallucinations. Below, we detail the datasets, evaluation metrics, baselines, and experimental setup.

### 3.1 Dataset

This study conducted a comprehensive evaluation on six challenging question-answering datasets: **HotpotQA** requires multi-hop reasoning over multiple documents. **NaturalQuestionsShort (NQ)** focuses on extracting short answers from real user search queries. **TriviaQA-web** uses open-domain trivia questions to test information retrieval from unstructured text. **SQuAD** is a classic reading comprehension dataset for evaluating paragraph-level answer extraction. **NQ-swap** is an adversarial test set with constructed context-knowledge conflicts to test model robustness. **NewsQA**, based on news articles, emphasizes the extraction of time-sensitive information. These datasets cover diverse task types and challenges, allowing for a multi-dimensional evaluation of the model's reasoning, knowledge integration, and information extraction capabilities.

### 3.2 Metrics

We evaluate model performance using two standard metrics in open-domain question answering: **Exact Match (EM)** and **F1 score**, both widely adopted in benchmarks such as SQuAD and NaturalQuestions.

Together, EM and F1 provide a complementary assessment: EM captures correctness at the string level, while F1 evaluates content overlap and fluency. This dual-metric approach allows us to distinguish between models that generate *precise* answers (high EM) versus those that produce *semantically similar but incomplete* outputs (high F1, low EM). For instance, in Table **??**, the greedy decoder outputs "Royce da 5'9" and E", achieving moderate F1 due to partial match, but fails EM due to incompleteness—whereas FreqAlign achieves both high EM and F1 by generating the full phrase.

While additional metrics such as BLEU or ROUGE could be considered, they are less suitable for QA tasks due to their sensitivity to surface form and lack of semantic grounding. We thus focus on EM and F1 as the most informative and interpretable measures for factual accuracy and content completeness.

### 3.3 Baselines

To validate the effectiveness of FreqAlign, this study compares it against four decoding strategies:

- **Greedy Decoding**: The standard approach of selecting the token with the highest probability at each step.
- **CAD** (Shi et al., 2023): A method that introduces richer contextual information and evidence dependencies to guide generation and ensure output consistency with the context.

- **COIECD** (Yuan et al., 2024): An adaptive decoding method that dynamically adjusts its generation strategy based on context and entropy constraints to identify and resolve knowledge conflicts.

- **DAGCD** (Huang et al., 2025b): A decoding framework that dynamically adjusts attention distribution and uncertainty signals to ensure generated content is contextually consistent.

- **FreqAlign (Ours)**: Our proposed method, which introduces token-level statistical frequency into the decoding process. It uses bidirectional frequency optimization to suppress globally common tokens and enhance contextually relevant ones. FreqAlign provides an interpretable, low-cost solution for context-faithful generation.

### 3.4 EXPERIMENTAL SETUP

The experiment was conducted on an NVIDIA H20-NVLink GPU cluster, with training and validation based on the LLaMA-2-7B model (Touvron et al., 2023). We extracted 2,000 samples from the HotpotQA training set using our frequency-weighted sampling method to train the Logistic Regression classifier. A five-fold cross-validation was used to optimize the regularization coefficient $C \in [10^{-4}, 10^3]$. The feature space combined signals from the top-10 most discriminative attention heads with the contextual word frequency. In the decoding stage, the maximum generation length was set to 10. The frequency fusion module integrated contextual and pre-computed global word frequency distributions with a weight ratio of 0.2:0.8.

Table 1: Performance comparison of different decoding methods on the same dataset. All baselines are reproduced under the same settings. Bold indicates the best performance. NQ denotes NaturalQuestionsShort.

| Methods | HotpotQA | | NQ | | TriviaQA–web | | SQuAD | | NQ-swap | | NewsQA | |
|---|---|---|---|---|---|---|---|---|---|---|---|---|
| | EM | F1 | EM | F1 | EM | F1 | EM | F1 | EM | F1 | EM | F1 |
| Greedy | 44.74 | 57.41 | 38.86 | 50.60 | 55.29 | 68.02 | 39.90 | 52.61 | 36.10 | 36.69 | 32.93 | 45.47 |
| CAD | 44.13 | 54.49 | 38.15 | 48.56 | 55.26 | 68.03 | 38.33 | 51.14 | 36.12 | 36.68 | 31.73 | 43.72 |
| COIECD | 42.03 | 51.48 | 38.78 | 51.68 | 57.05 | 70.04 | 40.93 | 54.79 | 38.79 | **51.68** | 34.98 | 35.61 |
| DAGCD | 45.21 | 55.01 | 44.12 | 55.73 | 56.75 | 68.38 | 46.92 | 58.81 | 48.25 | 48.91 | 35.49 | 47.41 |
| **FreqAlign** | **47.16** | **57.41** | **46.59** | **58.75** | **60.23** | **70.34** | **47.31** | **59.45** | **49.33** | 49.95 | **36.09** | **47.52** |

### 3.5 MAIN RESULTS

Table 1 presents a comparison of various decoding strategies across six QA datasets. The results show that FreqAlign consistently outperforms other methods, including greedy decoding, CAD, COIECD, and DAGCD, achieving significant improvements in both Exact Match (EM) and F1 scores. While standard greedy decoding shows reasonable performance on simpler tasks like TriviaQA-web, its effectiveness diminishes in more complex scenarios. Other advanced decoding methods like CAD, COIECD, and DAGCD offer moderate improvements. However, FreqAlign, with its dynamic adjustment of token probabilities through frequency alignment, consistently achieves the best or near-best performance across all datasets. For instance, on the NQ dataset, FreqAlign surpasses the strong DAGCD baseline by 2.47 points in EM and 3.02 points in F1 score. This demonstrates the effectiveness of our frequency-based approach in enhancing the factual accuracy and semantic coherence of the generated answers.

### 3.6 DATASET-SPECIFIC ADVANTAGE ANALYSIS

The consistent superiority of FreqAlign across six diverse datasets can be attributed to its frequency-aware calibration mechanism, which adapts to the lexical and structural characteristics of each task. Below, we provide dataset-specific explanations grounded in our algorithm's mathematical formulation.

**HotpotQA & TriviaQA-web**: These datasets require multi-hop reasoning or retrieval from noisy web text. FreqAlign's *global frequency debiasing* (Eq. 4) suppresses generic connector words (e.g., "the", "and") that dominate statistical priors but carry low semantic value. This allows rarer,

context-critical entities (e.g., "Zaheer Khan", "1978") to rise in rank — precisely what multi-hop QA demands.

**NaturalQuestionsShort (NQ) & SQuAD**: These datasets contain short, factoid answers often composed of low-frequency tokens (dates, numbers, names). Our *contextual frequency enhancement* (Eq. 5–6) boosts tokens like "1.95 m" or "7 October" by normalizing their occurrence within the local context. Since these tokens appear repeatedly in their respective passages, $f_{ctx}(w)$ assigns them high relevance scores, directly countering LLMs' tendency to truncate or substitute them.

**NQ-swap**: This adversarial dataset contains context-knowledge conflicts. FreqAlign's classifier, trained on attention-frequency joint features, learns to down-weight tokens that are globally frequent but locally inconsistent (e.g., a swapped entity name). The logistic regression score $s(w) = \mathbf{w}^{*\mathrm{T}}\mathbf{v}$ acts as a gatekeeper, filtering out statistically plausible but contextually invalid tokens.

**NewsQA**: Answers often involve time-sensitive entities and measurements. The bidirectional alignment ensures that even if "October 7" has low global frequency, its high contextual recurrence (e.g., appearing in datelines or event summaries) elevates its generation probability. This explains FreqAlign's +0.61 F1 gain over DAGCD — a critical improvement for temporal fidelity.

In all cases, the core mathematical advantage lies in Eq. (3): $\mathrm{combined}_{\mathrm{freq}} = \gamma \cdot f_{ctx}(w) + (1 - \gamma) \cdot f_{\mathrm{global}}(w)$. This linear interpolation allows FreqAlign to dynamically adapt its bias-correction strength per dataset — suppressing noise in web text (high $1 - \gamma$) while amplifying signal in dense contexts (high $\gamma$).

### 3.7 EFFICIENCY ANALYSIS

The FreqAlign mechanism optimizes the token generation process by dynamically adjusting selection probabilities during inference.

**Frequency-attention synergy**: During the training of the classifier, frequency information is incorporated with attention signals. The classifier then informs the decoding-time adjustments. During generation, the probability distribution is adjusted at each step based on the generated tokens and context. The time complexity for generation is approximately $O(nL)$, where $n$ is the maximum generation length and $L$ is the number of model layers. The space complexity for storing frequency distributions and attention weights is $O(nL)$.

**Contextual frequency enhancement**: FreqAlign adjusts token probabilities based on global and contextual frequency distributions. For each input of length $n$, calculating the frequency distribution has a time complexity of $O(n)$, and storing this information requires $O(n)$ space. This process improves the semantic accuracy of the generated content.

**Reduced hallucinations**: By adjusting the probability distribution during generation, the model can reduce the generation of irrelevant or factually incorrect information. This improves overall efficiency by reducing the need for post-processing corrections and preventing error accumulation.

**Probability-based threshold control**: By using a threshold system for attention and frequency in the classifier, only tokens with high scores on both metrics are considered highly relevant. This helps focus computational resources on the most likely correct answers.

The combination of contextual and global frequency adjustments improves efficiency by enhancing prediction accuracy, reducing irrelevant information, and minimizing the computational cost of processing unimportant data. This results in a more accurate, faster, and robust system.

### 3.8 ABLATION STUDY

**Variant 1: Detector training data sizes.** We evaluated the impact of the training data size on the performance of our context-aware classifier. As shown in Figure 2, our approach is data-efficient, achieving strong results even with as few as 100 training samples and showing consistent performance across different data scales.

**Variant 2: Exploring bidirectional word frequency.** We conducted an ablation study to understand the contribution of the global and contextual frequency components. The results are shown in Table 2.

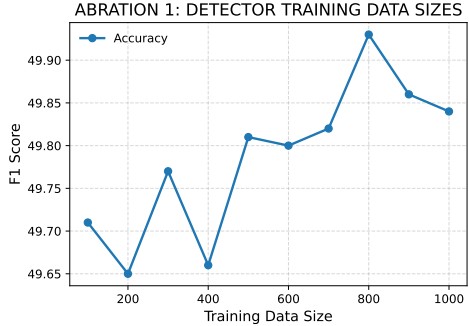

Figure 2: Results of FreqAlign with Different Training Data Sizes

Table 2: Results of ablation study. Here GWF and CWF represent global word frequency and context word frequency, respectively.

| Variants | EM | F1 |
|---|---|---|
| FreqAlign (full version) | **47.16** | **57.41** |
| FreqAlign w/o GWF | 46.52 | 56.89 |
| FreqAlign w/o CWF | 46.85 | 57.04 |
| FreqAlign w/o GWF and CWF | 45.13 | 54.84 |

- Removing global word frequency (GWF) led to a drop of 0.69 in EM and 0.52 in F1. This suggests that penalizing globally common tokens is effective at removing high-frequency but useless words from the top of the distribution, thereby improving accuracy.

- Removing contextual word frequency (CWF) resulted in a smaller performance drop. This is likely because the context-aware classifier, which is trained with contextual frequency signals, already enables the model to identify and reinforce evidence-backed tokens to some extent.

- Removing both GWF and CWF caused a significant performance decline, especially a 2.57 drop in F1 score. This highlights that the full bidirectional frequency mechanism is crucial for improving the model's ability to generate semantically relevant and complete answers.

**Variant 3: Bidirectional word frequency fusion ratio.** We experimented with different fusion ratios for combining the global and contextual frequency signals. As shown in Figure 3, the model's performance peaks when the contextual word frequency proportion is set to 20%. While performance varies with other ratios, all configurations still outperform the greedy baseline, confirming the general effectiveness of the FreqAlign approach.

### 3.9 CASE STUDY

To illustrate FreqAlign's behavior, we present a representative case. Additional cases demonstrating attribute extraction and structural integrity preservation are provided in Appendix A.

**Analysis of Case No.1:** The results are shown in Table 3. When generating date entities, standard models can be prone to premature termination or fall into generic patterns because the correct date tokens do not receive sufficiently high probabilities. The greedy approach is interrupted before completing the date. FreqAlign addresses this by using global debiasing to lower the probability of high-frequency but irrelevant tokens (e.g., "on", "born"), creating space for the correct tokens. Simultaneously, it enhances the probability of contextually relevant tokens like "7", "October", and "1978", enabling the model to correctly select them and form the complete date.

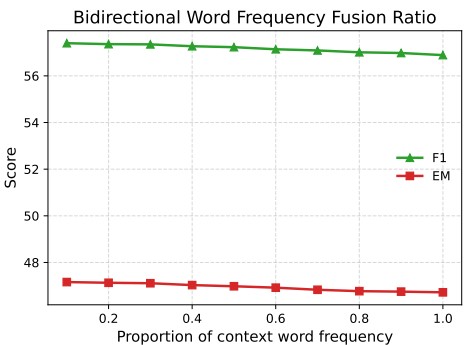

Figure 3: Results of FreqAlign with Different Fusion Ratio.

Table 3: FreqAlign improves date generation by reducing irrelevant token probabilities and enhancing context-specific date token selection.

| Question | When was the former Indian cricketer who made his ODI debuts during the 2000 ICC KnockOut Trophy born? |
| --- | --- |
| Context Key | Zaheer Khan (born 7 October 1978) |
| **Golden Answer** | **7 October 1978** |
| Greedy Answer | Zaheer Khan was born on |
| **FreqAlign Answer** | **7 October 1978** |

## 4 CONCLUSION AND FUTURE WORK

This paper introduces FreqAlign, a novel approach to mitigate context hallucinations in LLMs by calibrating token probability distributions through bidirectional frequency alignment. It combines global frequency debiasing (suppressing generic, high-frequency tokens) with contextual frequency enhancement (boosting relevant, potentially low-frequency tokens). Evaluation on six QA benchmarks demonstrates that FreqAlign significantly outperforms existing state-of-the-art baselines, improving both factual accuracy (EM) and semantic consistency (F1), thereby enhancing the reliability of generated text. In future work, we plan to: 1) explore the integration of character-level, subword-level, or multimodal frequency signals and 2) develop dynamic mechanisms to adjust the balance between global and contextual frequency influence based on task or input complexity.

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

# A  ADDITIONAL CASE STUDIES

**Analysis of Case No.2:** This case demonstrates FreqAlign's ability to extract attributes that are distant from the main entity in the text. The height information ("1.95 m") is separated from the player's name by 12 tokens and competing numerical information (weight). The greedy model falls into a common cognitive trap, answering with the most prominent entity ("Jesse Hogan"). FreqAlign, however, enhances the semantic link between the question ("height") and context ("tall"), while its frequency alignment breaks the statistical bias that a person's name is always the most likely answer. This allows it to correctly extract the attribute and its unit, achieving a complete inference from entity recognition to attribute parsing.

Table 4: FreqAlign improves attribute extraction by strengthening associations, thereby overcoming cognitive biases in measurement-related tasks.

| Question | What is the height of the player who won the 2015 AFL Rising Star award? |
| --- | --- |
| Context Key | Jesse Hogan (born 12 February 1995) is a professional Australian rules footballer playing for the Melbourne Football Club in the Australian Football League (AFL). A key forward, Hogan is 1.95 m tall and weighs 100 kg. |
| **Golden Answer** | **1.95 m** |
| Greedy Answer | Jesse Hogan |
| **FreqAlign Answer** | **1.95 m** |

**Analysis of Case No.3:** In this example, the greedy model produces an incomplete answer, likely because it tends to output simpler structures and may consider content within parentheses as secondary information with lower probability. FreqAlign counteracts this tendency toward "over-simplification." By enhancing the probability of the precise string found in the context—"Royce da 5'9" (Bad) and Eminem (Evil)"—it encourages the model to output the complete, original expression. This demonstrates how FreqAlign can improve the structural integrity of the generated output by resisting abbreviation biases and adhering more closely to the source text.

Table 5: FreqAlign ensures structural integrity by preventing model simplification and enhancing the likelihood of complete, contextually accurate outputs.

| Question | What are the names of the members of the Detroit-based hip hop duo who has worked with Jason Gilbert? |
| --- | --- |
| Context Key | Bad Meets Evil is an American hip hop duo composed of Detroit-based rappers, Royce da 5'9" (Bad) and Eminem (Evil). |
| **Golden Answer** | **Royce da 5'9" (Bad) and Eminem (Evil)** |
| Greedy Answer | Royce da 5'9" and E |
| **FreqAlign Answer** | **Royce da 5'9" (Bad) and Eminem (Evil)** |

