# OpenReview forum: "FreqAlign: Frequency-Based Calibration for Mitigating Contextual Hallucinations in Large Language Models"
_ICLR.cc/2026/Conference — ICLR 2026 Conference Withdrawn Submission_

### Official Review · Reviewer_SVxB · 2025-10-28

**Soundness:** 3
**Presentation:** 3
**Contribution:** 3
**Rating:** 6
**Confidence:** 3

**Summary:**

The paper introduces FreqAlign, a frequency-based calibration method designed to reduce contextual hallucinations in large language models. The method works by realigning token probability distributions through bidirectional frequency alignment: it suppresses globally frequent but contextually irrelevant tokens and boosts contextually important but infrequent ones. The approach uses a logistic regression classifier trained on attention-weight and frequency features to guide decoding-time adjustments. Evaluations across six QA benchmarks (HotpotQA, NaturalQuestionsShort, TriviaQA-web, SQuAD, NQ-swap, NewsQA) demonstrate consistent improvements over strong baselines like DAGCD, CAD, and COIECD.

**Strengths:**

1. Novel perspective: Introduces a simple yet effective method to leverage token frequency information for mitigating hallucinations, which is rarely explored systematically in prior work.
2. Decoding-time efficiency: Operates at inference time without requiring model retraining, making it compatible with existing LLMs.
3. Comprehensive experiments: Evaluated on diverse QA datasets with consistent improvements in both EM and F1 metrics.
4. Interpretability: The method is inherently interpretable — frequency-based recalibration intuitively links to linguistic patterns and contextual grounding.
5. Ablation and case studies: Ablation experiments and qualitative examples (e.g., date completion, numerical precision, entity disambiguation) demonstrate clear, explainable benefits of the approach.

**Weaknesses:**

1. Simplicity vs. novelty: While effective, the technique is conceptually straightforward and builds on established ideas in decoding adjustment and lexical bias correction. The novelty may be considered incremental for ICLR standards.
2. Limited theoretical depth: The paper lacks formal justification or probabilistic analysis of why frequency alignment systematically improves factual consistency.
3. Restricted evaluation scope: All evaluations are on QA datasets; results on summarization, dialogue, or instruction-following tasks would strengthen generality claims.
4. Classifier design limitations: The classifier is based on logistic regression using attention and frequency features — while simple, it may underutilize rich contextual representations.
5. Trade-off analysis missing: The paper doesn’t quantify how much frequency calibration affects fluency, diversity, or creativity compared to hallucination mitigation.
6. Hyperparameter sensitivity: The frequency fusion ratio (γ) and weighting factors are chosen empirically. More discussion on stability or automatic tuning would add credibility.

**Questions:**

1. How does FreqAlign perform under higher-generation temperature or sampling-based decoding methods?
2. Is the classifier generalizable across models (e.g., from LLaMA-2 to Mistral or Gemma)?
3. Does the method improve factual grounding in tasks beyond QA, such as summarization or reasoning benchmarks?
4. Could neural frequency models replace the simple logistic regression classifier for more adaptive calibration?
5. How sensitive are results to the frequency corpus (e.g., domain-specific corpora for global frequencies)?

---

### Official Review · Reviewer_cUo7 · 2025-10-30

**Soundness:** 2
**Presentation:** 2
**Contribution:** 2
**Rating:** 2
**Confidence:** 4

**Summary:**

This paper addresses the context hallucination in large language models (LLMs), where models generate information that contradicts or fabricates evidence from the retrieved context. The authors propose FreqAlign, a frequency alignment-based framework designed to mitigate such hallucinations. The authors first construct positive and negative samples based on their contextual influence, and then train a context-aware classifier that measures contextual relevance between context and potential answers.
After that, this classifier is used to recalibrate the importance distribution of input tokens through frequency alignment with the dual adjustment mechanism, which suppresses globally frequent tokens likely to induce hallucinations while amplifying contextually relevant but infrequent tokens. Experimental evaluations on six standard question-answering benchmarks show consistent and significant improvements over strong state-of-the-art baselines. The authors also release the full implementation and training setup to promote reproducibility.

**Strengths:**

1. The research problem is important, as mitigating LLM hallucinations has significant implications for improving model reliability and real-world applicability.

2. The motivation is reasonable and clearly presented, and the proposed method is conceptually simple.

3. The paper is easy to follow, with a clear structure and straightforward exposition.

**Weaknesses:**

1. The effectiveness of the method is questionable. According to Lines 105–106, the authors train a classifier to predict the relevance between contextual tokens and answers. This is a very difficult task, as the training corpus and model capacity of the classifier are far less than the LLM itself, and this paper tries to utilize a much weaker model to enhance a stronger backbone LLM. The methodologiy and mechanism are under-explained; more discussion is needed.

2. The method lacks theoretical insight. The overall design and parameter settings rely heavily on handcrafted priors. Since the paper is submitted under the Optimization primary area, more theoretical or analytical justification would be expected.

3. The experimental conclusion appears unconvincing. The chosen datasets are relatively simple. How does the method perform on more challenging benchmarks such as MME or LongBench? What about tasks requiring more complex reasoning? Additionally, performance is only reported on Llama2, and results on other model families should be included for completeness.

**Questions:**

1. Figure 1 is poorly presented. It is unclear how the positive/negative samples are constructed and how the classifier is actually applied in the pipeline.

2. In Eq. (2), frequency is computed within the current context. Wouldn’t that make most token frequencies roughly similar in the same context? How does the model distinguish token importance in this case? Why not compute frequency from the entire corpus instead?

3. In Eq. (2), why is the denominator the maximum frequency rather than the sum?

4. What is the architecture of the classifier? Is it just a single linear layer? If so, can it really capture such complex contextual relationships?

---

### Official Review · Reviewer_aF3K · 2025-10-31

**Soundness:** 1
**Presentation:** 3
**Contribution:** 2
**Rating:** 2
**Confidence:** 3

**Summary:**

This work focuses on a critical challenge in large language models (LLMs): contextual hallucinations, where generated outputs contradict or fabricate information relative to the provided context.
The authors propose FreqAlign, a novel frequency alignment-based calibration method designed to mitigate such errors.
FreqAlign introduces a dual-stage approach.
First, it trains a context-aware classifier using token frequency information derived from positive/negative sample pairs based on actual contextual influence.
Second, this classifier recalibrates token importance by aligning frequencies, specifically suppressing globally frequent tokens likely to cause hallucination while boosting semantically important but infrequent ones.
Experimental results show the improvements from FreqAlign.

**Strengths:**

1. This wor articulates why global token frequency can be problematic (e.g., overused generic words may induce hallucinations) and how focusing on contextually salient yet rare tokens can improve factuality.
2. The authors leverage a simple yet effective statistical cues grounded in token usage patterns within contexts.

**Weaknesses:**

The major concern lies in the experiments.
It seems that all the results and analysis are made on a single LLaMA-2-7B model.
However, for a calibration method, it is important to show the generalizability across various backbone models, especially on those larger and more powerful models which have higher performances.
Make improvements on just one less powerful model could not provide enough soundness to support the effectiveness of the proposed methodology.

Typo: Incorrect table index reference on line 254.

**Questions:**

Is there any experimental evidence to show the improvement on other models?

---

### Official Review · Reviewer_q1M4 · 2025-10-31

**Soundness:** 2
**Presentation:** 2
**Contribution:** 2
**Rating:** 2
**Confidence:** 4

**Summary:**

This paper introduces a method named FreqAlign, which can be used to reduce hallucinations in Large Language Models. The method proposed in the paper functions by recalibrating the token probability distribution through a frequency-based approach, namely: 1) suppressing generic or high-frequency words across a corpus in Global Frequency De-biasing, and 2) amplifying the probability of tokens that are frequent within the provided input context in Contextual Frequency Enhancement. The author trained a classifier on attention weights and token frequencies. The classifier is later used to guide this probability adjustment procedure. Based on their empirically finding, the method show decent performance of the method on both Exact Match and F1 scores for different datasets.

**Strengths:**

1. The key idea of this paper is to use frequency analysis to calibrate LLM outputs for reduction of hallucination, which is interesting. The method is built upon the observations that  LLM tend to fall back to high-frequency, generic language rather than contextually specific and rarer tokens.
2. The dual mechanism suppresses common words from global context and amplifies the locally relevant ones in an ingenious and direct way to resolve this misalignment of probability in the output logits.
3. The proposed method is easy to follow with the help me of figures,

**Weaknesses:**

1. The balance between global and contextual frequency, regulated by the fusion ratio, was fixed across all experiments. While it yielded the good performance, my concern is that this single setting is universally optimal across domains and contexts. A more sophisticated model that with dynamically adapts based on context properties, could have better results.
2. The experiments are limited to a single model, LLaMA-2-7B, and only question-answering tasks. It needs to investigate whether these results generalize to other models from other architectural or families. My concern is that if It can be applied to more abstractive, longer-form generation tasks where hallucinations may be harder to detect. This paper also ignore many baseline methods such as DoLA.
3. The reliance on frequency will have some downsides. For instance, a very common case where a correct answer token that occurs only once in a long context may get an insufficient boost from the contextual frequency enhancement. On the other hand, an incorrect entity may be mistakenly amplified if it is repeated frequently in a document with a lot noise.
4. There is no limitation section in the paper. Besides, there should be more ablation study. For example, the final results should depend on the accuracy of the trained-classifier. If the classifier has low accuracy, the final results can be totally wrong. The paper also lack of discussion on its latencey, which could make the method itself impractical

**Questions:**

see weakness

---

### Note · Authors · 2026-03-25

**Comment:**

Withdrawn based on the reviewers' comments.

**Withdrawal Confirmation:**

I have read and agree with the venue's withdrawal policy on behalf of myself and my co-authors.

---

### Meta-Review · Area_Chair_MVXZ · 2025-12-07

**Summary:**

Three reviewers recommend rejection, highlighting concerns regarding the paper’s technical novelty and empirical validation. Additional issues include limited experimental results, missing comparisons to relevant baseline methods, insufficient ablation analysis, and poor presentation and clarity.
The authors did not submit a rebuttal, leaving these concerns unaddressed. Therefore, I recommend rejection.

**Reviewer Concerns:**

The rebuttal was not submitted.

**Reviewer Scores:**

The rebuttal was not submitted and therefore did not affect the reviewers initial decisions.

---

### Decision · Program_Chairs · 2026-01-26

Reject